# Bioinspired Nanoplatforms Based on Graphene Oxide and Neurotrophin-Mimicking Peptides

**DOI:** 10.3390/membranes13050489

**Published:** 2023-04-30

**Authors:** Luigi Redigolo, Vanessa Sanfilippo, Diego La Mendola, Giuseppe Forte, Cristina Satriano

**Affiliations:** 1Nano Hybrid Biointerfaces Lab (NHBIL), Department of Chemical Sciences, University of Catania, Viale Andrea Doria, 6, 95125 Catania, Italy; luigi.redigolo@studium.unict.it (L.R.); sanfilippo.vanessa@studium.unict.it (V.S.); 2Department of Pharmacy, University of Pisa, Via Bonanno Pisano, 6, 56126 Pisa, Italy; diego.lamendola@unipi.it; 3Department of Drug and Health Science, University of Catania, Viale Andrea Doria, 6, 95125 Catania, Italy; giuseppe.forte@unict.it

**Keywords:** supported lipid bilayers, fluorescence recovery after photobleaching (FRAP), fluorescence resonance energy transfer (FRET), angiogenesis, molecular dynamics, confocal microscopy, atomic force microscopy, peptides

## Abstract

Neurotrophins (NTs), which are crucial for the functioning of the nervous system, are also known to regulate vascularization. Graphene-based materials may drive neural growth and differentiation, and, thus, have great potential in regenerative medicine. In this work, we scrutinized the nano–biointerface between the cell membrane and hybrids made of neurotrophin-mimicking peptides and graphene oxide (GO) assemblies (pep−GO), to exploit their potential in theranostics (i.e., therapy and imaging/diagnostics) for targeting neurodegenerative diseases (ND) as well as angiogenesis. The pep−GO systems were assembled via spontaneous physisorption onto GO nanosheets of the peptide sequences BDNF(1-12), NT3(1-13), and NGF(1-14), mimicking the brain-derived neurotrophic factor (BDNF), the neurotrophin 3 (NT3), and the nerve growth factor (NGF), respectively. The interaction of pep−GO nanoplatforms at the biointerface with artificial cell membranes was scrutinized both in 3D and 2D by utilizing model phospholipids self-assembled as small unilamellar vesicles (SUVs) or planar-supported lipid bilayers (SLBs), respectively. The experimental studies were paralleled via molecular dynamics (MD) computational analyses. Proof-of-work in vitro cellular experiments with undifferentiated neuroblastoma (SH-SY5Y), neuron-like, differentiated neuroblastoma (dSH-SY5Y), and human umbilical vein endothelial cells (HUVECs) were carried out to shed light on the capability of the pep−GO nanoplatforms to stimulate the neurite outgrowth as well as tubulogenesis and cell migration.

## 1. Introduction

Graphene and its derivatives have attracted tremendous interest over the past decade from scientists in the biomedical field due to many unprecedented intrinsic biological activity properties, in addition to their other unique electronic, optical, mechanical, and chemical properties. As of today, different bidimensional nanomaterials have been exploited for the repair of the injured neural cell. Graphene-family materials, thanks to their conductivity, are especially appropriate to prompt the regeneration of excitatory neurons and the formation of synapses. Graphene oxide (GO)-based nanomaterials can be used as drug delivery systems to aid the functioning of drugs for neural regeneration [1] and for neural differentiation of stem cells [2]. As an example, graphene surfaces have also shown enhancement properties on the branching and growth of neuronal circuits in hippocampal cells in mice [3]. Due to its strong non-covalent binding ability, reduced graphene oxide (rGO) can promote neuronal differentiation and myelination, while also enhancing the adhesion and osteogenic differentiation of bone marrow mesenchymal stem cells [4]. The neuron response to GO is size-dependent: GO flakes with lateral dimensions larger than 10 μm can be toxic and reduce neuronal viability [5], while flakes of less than 0.5 μm in size do not affect cell viability [6]. As a rule of thumb, the toxicity of graphene derivatives is highly dependent on their composition and chemical functionalization [7,8] as well as their size.

Neurotrophins (NTs), including the nerve growth factor (NGF), the brain-derived growth factor (BDNF), and neurotrophin 3 (NT-3), are secreted proteins required for maintenance, development, and differentiation of neurons [9,10,11,12]. They exert their function as homodimers activating two different membrane receptors: the p75 NT receptor, which is common to all NTs, and the tropomyosin receptor kinase (Trk) family that selectively recognizes different NTs (TrkA for NGF, TrkB for BDNF, and TrkC for NT-3) [13]. NTs are essential for synaptic plasticity and to modulate neuronal biochemical pathways that underlie memory formation as the expression of cAMP Response Element-Binding Protein (CREB) [14,15,16]. Indeed, the alteration of the activity and levels of neurotrophic factors are related to cognitive deficits and neurodegenerative diseases [17,18]. For this reason, treatments that modulate neurotrophin levels or direct administration of NTs have acquired a great deal of interest in preventing neurodegeneration, promoting neural regeneration in several pathologies [17,19,20,21] and results promising for the treatment of spinal cord injury and cerebral ischemia [22,23].

Recently, research has focused on NTFs as potential regenerative therapy for neurodegenerative diseases [24] and other related physio-pathological processes, such as angiogenesis, namely, the formation of new blood vessels from pre-existing ones. Indeed, NTs are expressed not only in the central nervous system and display several activities but also in non-neuronal cells [25,26,27]. All NTs are expressed in the epidermis and modulate the cutaneous microvascular network [28]. A piece of strong evidence has emerged that this family of proteins exerts an essential role in angiogenesis [29,30]. Endothelial cells express Trk receptors, while NGF can promote the proliferation of human umbilical vein endothelial (HUVECs) [31], human dermal microvascular endothelial [32], and rat brain endothelial cells [33]. NGF can induce corneal epithelial wound healing [34], to remodel skeletal muscle fibers in ischemic limbs [35], as well as NT-3 [36]. More recently, it has been demonstrated that NGF is implicated in the various stages of the burn wound healing process [37]. BDNF promotes endothelial cell survival and angiogenic tube formation, and induces neo angiogenesis in ischemic tissues [38,39]. NTs can also induce angiogenesis by stimulating and regulating the expression of angiogenic factor as vascular endothelial growth factor (VEGF) [40]. BDNF induces in the HUVEC the secretion of angiogenin a potent angiogenic factor able also to increase wound healing [41,42,43]. Therefore, a dynamic relationship between vascular and neuronal tissues exists: HUVECs secrete NTs, which in turn drive and enhance axonal growth in the peripheral nerve [44].

However, as potential drugs, neurotrophins show some drawbacks, such as low serum stability, poor oral bioavailability, the necessity to cross the blood–brain barrier (BBB), and the pain that their actual administration techniques cause [45,46]. Due to these problems, the therapeutic use of neurotrophins is difficult. It is, thus, important to find a way to smoothen the administration pathways of these drugs; the synthesis of peptides able to mimic specific domains of the NTs may represent an alternative to bypass the aforementioned shortcomings [47,48,49]. Considering the role of the N-terminal region as the key domain of NTs for the binding selectivity and activation of Trks [50,51,52], some of us have synthesized linear peptides encompassing the 1–14 residues of NGF (NGF 1-14) [53,54], the 1–12 residues of BDNF (BDNF 1-12) [55,56], and the 1–13 residues of NT-3 (NT3(1-13) [57] that displayed activity in the neurite outgrowth and CREB activation [58,59].

According to the above, following the concept of ‘adaptive biomaterials carried with neurotrophic factors’ [60], in this work, we exploited the ability of the three peptide sequences—BDNF(1-12)-FAM, NT3(1-13)-FAM, and NGF(1-14)-FAM, where -FAM stands for 5(6)-carboxyfluorescein, which is the fluorescent dye used to label the peptides via covalent bonds and immobilized onto GO sheets via physisorption—to promote angiogenesis and neurogenesis. Indeed, GO also has a great potential for applications in angiogenesis, as well as cancer therapy, due to its pro- or anti-angiogenic activity and wound healing potential [61]. Indeed, GO and rGO have been extensively researched because the modulation of defects in graphene-based nanomaterials is of direct relevance for a variety of applications in biomedical engineering, such as drug delivery, biosensing, cancer therapeutics, and tissue engineering [62]. GO has been recently demonstrated to act as a regulator of apoptosis and autophagy, which are the two major forms of programmed cell death in both normal and cancer cells [63].

The interaction between the peptide-carrying nanoplatforms and an artificial model of the cell membrane was investigated, both experimentally and theoretically, using supported lipid bilayers (SLBs), which were formed via the absorption–rupture–fusion process of lipidic small unilamellar vesicles (SUVs) on ozonized hydrophilic silica substrates [64] (see representation in Figure 1). Graphene nanosheets and GO are known to work in opposite directions in the perturbation of the phospholipid membrane, with inserted graphene leading to a decreased lipid flip-flop rate, while embedded GO can catalyze the transport of phospholipids between membrane leaflets through facilitating the formation of water pores [65].

In vitro cellular experiments were carried out with undifferentiated (SH-SY5Y) and differentiated (dSH-SY5Y) neuroblastoma cells, to assess the effect of the pep−GO nanoplatforms in terms of neurites’ length and branching points, and on endothelial cells (HUVEC), to analyze the cellular response in terms of angiogenesis.

## 2. Materials and Methods

### 2.1. Materials and Reagents

The lipids 1,2-dipalmitoyl-sn-glycerol-3-phosphoethanolamine-N-(lissamine rhodamine B sulfonyl) (PE-Rhod) and 1-palmitoyl-2-oleoyl-sn-glycerol-3-phosphocholine (POPC) were purchased from Avanti Polar Lipids (Alabaster, AL, USA). The solvents chloroform and ethanol were purchased from Merck Millipore (Burlington, MA, USA). Graphene oxide (GO) was purchased from Graphenea Inc., Cambridge, MA, USA; as reported by the producer, sulfur and nitrogen contents were, respectively, below 3% and 1% in composition. The phosphate buffer saline (PBS) tablets were purchased from Sigma-Aldrich (St. Louis, MO, USA). All experiments were carried out with ultrapure water (18.2 mΩ·cm at 25 °C, total organic carbon (TOC) lower than 5 parts per billion (ppb), Ultrapure Millipore^®^ Water Type, Burlington, MA, USA). Dulbecco’s modified eagle medium (DMEM)-F12, DMEM high glucose, penicillin-streptomycin solution, L-glutamine, fetal bovine serum (FBS), Dulbecco’s phosphate-buffered saline (PBS), and retinoic acid (RA) were purchased from Sigma-Aldrich (St. Louis, MO, USA).

### 2.2. Preparation of Self-Assembled Lipids (SUVs, SLBs)

Rhodamine-labeled POPC SUVs, with a diameter of about 100 nm, were synthesized using POPC and PE-Rhod lipid chloroform solutions. Briefly, POPC at the initial concentration of 25 mg/mL was mixed with PE-Rhod at the initial concentration of 1 mg/mL in chloroform in a round bottom flask to obtain, in a volume of 1 mL of chloroform, phospholipid vesicles at the final concentration of 5 mg/mL. Due to the light sensitivity of the dye-labeled system, the flask remained covered under an aluminum foil for the whole preparation procedure, from the initial steps to the final storage.

The chloroform was evaporated inside a cabinet under N_2_ flux; to obtain a uniformly deposited dry lipidic film, during the evaporation time, the flask was maintained under a constant and wide radius rotative motion. Residual chloroform was removed by leaving the flask for 30 min in a rotating evaporator at room temperature; in the end, the lipidic film was hydrated with filtered and degassed PBS, 10 mM at pH = 7.4; controlling the pH was, indeed, necessary to simulate the physiological conditions [66].

Before the extrusion of the vesicles to fabricate SUVs, the syringes and cylinders composing the extrusion apparatus were washed, in order, in ethanol, Milli-Q H_2_O, and PBS (the latter only for the syringes); water was deionized and purified using a Milli-Q unit. The first extrusion step was performed with a 100 nm filter, while a second, finer one was performed with a 30 nm filter: the extrusion process was, in both cases, repeated an odd number of times, 13, to ensure the extruded SUVs ended in the originally empty syringe. The final product was stored, repaired from sunlight at 5 °C in a vial, insufflated with N_2_, and sealed with parafilm.

The SLB was formed via the vesicle absorption–rupture–fusion process [67] on glass surfaces (22 mm glass bottom dishes, Willco Well), which were ozone-treated to increase their hydrophilic character. This treatment required the dishes to be initially washed with ultra-pure Milli-Q water and treated two times in the ozonizer for 15 min, washing them with water after each step to clean the surface from the easily removable oxidized organic species. After the last rinsing, the SUVs were rapidly deposited on the dishes, using a micropipette to repeatedly aspire and deposit the vesicle solution, to mechanically aid the breaking and spreading process on the bottom of the dish, and left in incubation for 30 min.

### 2.3. Peptides Synthesis and Purification

The peptide sequences NT3(1-13), NGF(1-14), and BDNF(1-12) were synthesized with amidated C-termini [55,57], and all amino acid residues were added according to the TBTU/HOBT/DIEA activation protocol for Fmoc chemistry on a NovaSyn-TGR resin. The purification of peptides was performed via preparative reverse-phase high-performance liquid chromatography (RP-HPLC), using a Varian PrepStar 200 model SD-1 chromatography system with a Prostar photodiode array detector, with detection set at λ = 222 nm. The characterization was carried out via analytical rp-HPLC and ESI-MS [55], and the results for molecular weights and the sequence for each peptide are listed in Table 1.

### 2.4. Peptides Immobilization on GO Sheets

Fluorescent peptides were immobilized on previously prepared sheets of base-washed GO, prepared as explained in [68], previously suspended in PBS (1 mg/mL), and ultrasonicated for 2 h. The GO dispersion was diluted in PBS to obtain 4 aliquots with the new concentration of 200 μg/mL, and subsequently mixed with 100 μM PBS solutions of the fluorescent peptides from Table 1 to obtain four 400 μL volume systems, with the final concentration of 100 μg/mL in GO and 50 μM in the respective peptide.

To remove the loosely bound molecules, two purification steps were performed in the centrifuge for 15 min at 8000 rpm (using AMICON tubes with 3 K molecular weight cut-off), washing the pellets with PBS after the first centrifugation; at the end, about 200 μL of each pellet were obtained.

### 2.5. Physicochemical Characterization

#### 2.5.1. UV-Visible (UV-Vis) Spectroscopy and Fluorescence Spectroscopy with Fluorescence Resonance Energy Transfer (FRET) Analysis

Absorption spectra for each system were recorded in the 200–700 nm λ range using a JASCO V-560 UV-vis spectrophotometer equipped with a 1 cm path-length cell.

Fluorescence spectra were recorded with a Cary Eclipse Fluorescence spectrophotometer, which had 0.5 nm resolution at room temperature. The spectra were collected using 5:2.5 nm slit widths for all measurements. The excitation wavelengths of 488 nm and 543 nm were used to excite the fluorescence of carboxyfluorescein (FAM) and rhodamine (Rhod), respectively. In the FRET experiments, the fluorescence spectra of the three pep−GO systems were compared with those of the free peptides in the first experiment and with the spectra obtained from the interaction between the SUVs and the pep−GO hybrids in the second experiment, which quantified the energy transfer process that happened in the two cases.

#### 2.5.2. Scanning Confocal Microscope (LSM) Imaging and Fluorescence Recovery after Photobleaching (FRAP) Analysis

Confocal microscopy studies were executed with an Olympus FV1000 confocal laser scanning microscope (LSM), which was equipped with diode UV (405 nm, 50 mW), multiline Argon (457 nm, 488 nm, 515 nm, total 30 mW), HeNe(G) (543 nm, 1 mW), and HeNe® (633 nm, 1 mW) lasers. An oil immersion objective (60xO PLAPO) and spectral filtering systems were used. A constant and fixed value was defined for the detector gain, and images were collected, in sequential mode, randomly all through the area of the well.

After the removal of 1 mL of PBS, 50× diluted solutions of pep−GO pellet dispersion were added to POPC-PE-Rhod SLBs to obtain the sample for the analysis. After an incubation time of 30 min, which was necessary to allow the pep−GO samples to interact with the SLBs, the systems were analyzed without further rinsing.

For the FRAP analysis, time-solved snapshots were acquired before and after the bleach was performed with an Ar laser at a high intensity (95% power), while scans of 256 × 256 pixels were collected with laser radiation at 488 and 543 nm and used to excite the FAM and Rhod fluorescence, respectively. An average of 5 spots per substrate were photobleached in each experiment, through translating the sample stage, with a circular region of interest (ROI) spot having a diameter of 20 μm. The FRAP image processing and analysis were performed on ImageJ 1.53a version (U. S. National Institutes of Health, Bethesda, MA, USA) utilizing the FRAP profiler macro. We normalized data to the initial (pre-photobleach) value, which enabled the percentages of photobleaching and fluorescence recovery within the laser region to be determined. For each sample, the emission recorded from the bleached spots was compared with that coming from contiguous non-bleached areas. The diffusion coefficients were calculated using Axelrod’s algorithm [69].

#### 2.5.3. Molecular Dynamics (MD)

MD simulations were performed at the approximation level of molecular mechanics using the Consistence Valence Force Field (CVFF) parametrization. The Smart Minimizer algorithm was used to perform a first geometry optimization, followed by an equilibration procedure. MD simulations were carried out under NVT conditions at 298 K. The interactions between the systems were studied in periodical conditions, in a 5.6 nm × 5.1 nm × 3.0 nm box, in the presence of the solvent water, which, for clarity, is not displayed in the images. All calculations were performed using the Biovia Material Studio 2017 package [70].

### 2.6. Biochemical/Cellular Characterization

#### 2.6.1. Cell Cultures and Maintenance

Neuroblastoma (SH-SY5Y line) and Human Umbilical Endothelial cells (HUVECs line) were cultured in complete medium (DMEM-F12 supplemented with 10% FBS, 2 mM glutamine, 100 U penicillin, and 0.1 mg/mL streptomycin for SH-SY5Y, and Medium 200 supplemented with 2% of Low Serum Growth Factor, LSGS, 2 mM glutamine, 100 U penicillin, and 0.1 mg/mL streptomycin for HUVECs). Cells were grown in tissue culture-treated Corning^®^ flasks T25 (Sigma-Aldrich, St. Louis, MO, USA) under a humidified atmosphere at 37 °C, with 5% CO_2_, in an incubator (Heraeus Hera Cell 150C incubator). To induce the differentiation, neuroblastoma SH-SY5Y cells were treated for 3 days with 10 μM of retinoic acid (RA) in DMEM high glucose supplemented with 1% FBS.

#### 2.6.2. Cytotoxicity

To perform the viability assay, SH-SY5Y cells and HUVECs were plated in a 96-well plates, at a density of 10^4^ cells per well, in their respective complete culture media, for 24 hrs (3 extra days in the case of differentiated SHSY5Y, see Section 2.6.1). Afterward, cells were treated for 24 h with the samples BDNF(1-2), NT3(1-13), NGF(1-14), GO, BDNF 1-12 + GO, NT3(1-13) + GO, NGF(1-14) + GO, at a concentration of 0.5, 1, and 2.5 µM for each peptide, and 1 µg/mL, 2 µg/mL, and 5 µg/mL for GO, for 24 h. Cytotoxicity was determined at 37 °C using the 3-(4,5-dimethylthiazol-2-yl)-2,5-diphenyltetrazolium bromide (MTT) at a concentration of 0.5 mg/mL After 3 h of incubation, the enzymatic reduction in MTT to the insoluble purple formazan product was detected through dissolving the crystals with 100 μL of dimethyl sulphoxide and, thus, measuring the absorbance at 570 nm using a Varioscan spectrophotometer. The experiments were performed in triplicate, and the results are presented as the means ± SEM. The statistical analysis was performed via Student’s *t*-test.

#### 2.6.3. Neurite Outgrowth

Neuroblastoma SH-SY5Y cells were plated in glass bottom dishes, with 22 mm of glass diameter at a density of 48 × 10^3^ cells per dish in complete medium DMEM-F12 and 10% FBS, for 24 h. Thereafter, both differentiated (see Section 2.6.1) and undifferentiated SH-SY5Y cells were treated with the diverse samples at a concentration of 1 µM for each peptide and 2 µg/mL for bwGO for 2 h. Optical bright field images were recorded with a Leica ICC50 W microscope immediately after the treatment (t = 0) and after 2 h of incubation and analyzed to assess the neurite outgrowth, using the NeuronJ (neurite tracing and quantification) plugin from the ImageJ Software (NIH, Bethesda, MD, USA).

#### 2.6.4. Wound Closure Assay

For the wound scratch assay, HUVECs cells were seeded at a density of 13 × 10^4^ and cultured in the complete medium until confluence into 48-well culture plates. For HUVECs seeding, the plates were previously coated with 2% *w*/*v* gelatin. The confluent cell monolayers were scratched and wounded using a universal sterile 10 μL pipette tip and then rinsed with the medium. Afterward, each well was treated with the different samples at a concentration of 1 µM for each peptide and 2 µg/mL for bwGO, in Medium 200 supplemented with 2% LSGS. Serial phase contrast images (Leica, Wetzlar, Germany) of the in vitro wounds were taken immediately after the treatment and after 3, 5, 7, 24, and 30 h of incubation, and the width of the separation wall was measured using the MRI Wound Healing Tool on the ImageJ software (version 1.50i, NIH).

#### 2.6.5. Tube Formation Assay

To perform the tube formation assay, the Matrigel matrix (Corning, NY, USA) was thawed overnight at 5 °C and spread over each well (130 μL) of a 48-well plate. The plate was then incubated for 30 min at 37 °C to allow the gel to solidify. Thereafter, HUVEC cells (35 × 10^3^) were seeded in 400 μL of Medium 200 with 2% LSGS and treated with the diverse samples at a concentration of 1 µM for each peptide and 2 µg/mL for bwGO. After 3 h of incubation at 37 °C, the tube structures were observed with a Leica microscope equipped with a digital camera, and three bright field images (magnification 4×) were captured for each sample. The formation of tube-like structures was quantified in terms of branching points with the ImageJ analysis software.

## 3. Results and Discussion

### 3.1. The Interaction of Peptide-Functionalized GO with Model Cell Membranes

We used the photoluminescence analysis to scrutinize the efficiency of energy transfer processes in the set of three pep−GO constructs at the interface with the lipid membrane.

The carboxyfluorescein (FAM) moiety and rhodamine (Rhod) fluorophore, anchored through covalent links to the peptide backbone and the lipid head (Figure 2), respectively, were utilized as the donor–acceptor pair in fluorescence resonance energy process (FRET) analyses.

#### 3.1.1. Three-Dimensional FRET Analyses

Figure 1 shows the fluorescence spectra recorded for the assembly of the pep−GO hybrids (Figure 1a) and their following interaction with the SUVs, i.e., the 3D artificial membranes model (Figure 1b). For both experiments, the occurrence of energy transfer processes can be figured out.

In particular, the intensity in the peptides’ FAM moiety emission is expected to decrease upon the interaction with GO, which is a known quencher of fluorescence [71]. This finding is especially evident for NT3(1-13) and GO and NGF(1-14) and GO samples, where the emission drops, respectively, to 12.5% and 43.3% of the value of the free peptide (Figure 1a); this trend is likely related to strong π−π interactions that induce energy transfer or non-radiative dipole–dipole coupling. Surprisingly, in the case of BDNF(1-12) and GO, the change in the emission intensity of the fluorophore is rather the opposite, with a 112% increase in emission compared to the free peptide. In this case, the peptide adsorption onto the GO nanosheets can contribute to reducing the molecular aggregation of the dye molecules (in turn, induced via π−π stacking or hydrophobic interactions), which is generally known to reduce the brightness of fluorophores [72]; thus, the self-quenching of the fluorophore is reduced. It is worth pointing out that a hypsochromic shift of 2–3 nm for the absorption of the FAM group bound to NT3(1-13) and NGF(1-14) peptides, and a shift of 5 nm for the case of BDNF(1-12), are evident. These findings suggest that different structural transitions may occur in regard to the FAM-bounded peptide backbones, owing to their assembling with the GO nanosheet substrates [73].

The FRET efficiency (E), which was calculated for the three pep−GO systems, is given in Table 2.

Energy transfer processes also occurred between the FAM-labelled peptide molecules immobilized at the surface of the GO sheets (FRET donor) and the Rhod-labelled lipid vesicles (FRET acceptor). Figure 1b shows, in general, that in all spectra the expected trend was for an effective donor-acceptor pair, i.e., the decrease in the emission peak of carboxyfluorescein ($F_{D}^{\prime }$) and the increase in that of the rhodamine ($F_{A}^{\prime }$) to the fluorescence of the fluorophores alone.

The quantitative analysis of the emission intensity changes for the three pep−GO systems interacting with the 3D model of cell membrane displayed in Table 3 suggests two different mechanisms of energy transfer and/or dipole–dipole interaction for the Rhod-labelled SUVs, with the two hybrids of FAM-labelled pep−GO nanoplatforms of BDNF(1-12)−GO and NGF(1-14)−GO on the one side, and two hybrids of NT3 (1-13)−GO on the other side.

For all three pep−GO constructs, the $F_{A}^{\prime }/F_{A}>1$ ratio points to the increase in the emission intensity of the acceptor moiety upon the interaction with the donor. The enhanced acceptor’s emission exhibits the following increasing order: NT3(1-13)−GO < NGF(1-14)−GO < BDNF(1-12)−GO. On the other hand, the efficiency of the FRET process displays a different sequence order: NT3(1-13)−GO < BDNF (1-12)−GO < NGF(1-14)−GO. These findings suggest that the photoluminescence properties at the pep−GO(FAM)/SUV(Rhod) biointerface are a result of a complex process where multiple energy transfer mechanisms may occur, in addition to the simple donor–acceptor pair model.

To shed light on the nature of the interaction at the pep−GO/lipid membrane interface, which can be of help to unravel such multiple mechanisms, computational analyses using MD were carried out. The simulation models built for the three pep−GO materials are shown in Figure 2, while the calculated energies of interaction, average orientation, and interaction forces at the peptide/GO interface are summarized in Table 4.

It is known, in the case of GO, that the interface with amino acids is mainly governed via electrostatic and π−π interactions [75]. Moreover, it is known that the main functional groups participating in H-bonding are tertiary alcohols, while epoxides play only a minor role [76]. For instance, the calculated average H-bond energies for GO in water range from approximately 5 to 15 kcal/mol [76].membranes-13-00489-t004_Table 4Table 4Comparison between energy, orientation [77], and kind of interaction for three pep−GO simulated systems.Pep−GO Interaction ForcesBDNF(1-12)Electrostatics, H-bonds between GO and NH (Arg, 2) and COOH (Asp, 1)NT3(1-13)More effective contact surface for H-bonds between GO and NH (Arg, 1) and COOH (Glu, 2)NGF(1-14)The highest number of binding sites between GO and NH (Arg, 1) and COOH (Glu, 1), lower H-bond strength.

Our results for BDNF(1-12)−GO (Figure 2a), with a calculated interaction energy of −64.04 kcal/mol compared to the non-interacting single constituents, indicate that the hybrid peptide-GO nanosheet material has good stability. Notably, the most important interactions appear to be electrostatic—mostly hydrogen bonds (H-bonds, 35.61 kcal/mol)—as they mainly take place between the GO surface and the arginine’s guanidine groups and the carboxylic group from aspartic acid residues, while the glutamic acid moiety appears to be highly distanced from the surface.

In the case of NT3(1-13)−GO (Figure 2b), the interaction energy of the peptide with the GO substrate in the adlayer formation is −67.75 kcal/mol, i.e., almost four units more stable regarding the value given through the BDNF(1-12) sequence. Finally, NGF(1-14)−GO (Figure 2c) displays the highest interaction energy of −68.54 kcal/mol, with major contributions through H-bond formation (42.75 kcal/mol) between the GO surface and, especially, the guanidine groups from the arginine moieties and glutamic acid’s carboxylic group. In conclusion, the NGF(1-14)−GO hybrid appears to be the most energetically stable system and the system with the most effective interaction between the peptide and the GO surface, i.e., the highest number of H-bonds. It shows a stability increment of 0.79 kcal/mol to the NT3(1-13)−GO interaction, which is, in turn, 3.71 kcal/mol more stable than the increment given through BDNF(1-12)−GO.

These results are in agreement with the literature, being known that in the interactions of GO with amino acids, the highest stability is observed in the case of tryptophan and arginine [75].

The comparison between the calculated temporal evolution of the H-bond length for guanidine and carboxylic groups interacting with the GO substrate for the pep−GO set is shown in Figure 3.

Concerning the H-bonds’ distance between guanidine groups and GO (Figure 3a–c, top panels), there is a similar H-bond length oscillation around a mean value of about 2 Å for all three pep−GO models.

On the other hand, the calculated H-bond distance between carboxylic groups and GO (Figure 3a–c, bottom panels) exhibits an oscillation of the mean values of around 2.0 Å for BDNF(1-12), 2.2 Å (with fluctuations reaching up to 2.6 Å) for NT3(1-13), and 1.5 Å (with larger fluctuations that can reach up to 2.5 Å or down to 1.2 Å) for NGF(1-14). These findings point to the strongest interaction between NGF(1-14) and the GO surface among those subjects tested in our MD calculations.

#### 3.1.2. Two-Dimensional FRAP Analyses

Figure 4 shows representative sequences of fluorescence micrographs collected for the 2D FRAP experiments and performed via LSM for the Rhod-labelled POPC SLB, before and after the interaction with the pep−GO systems.

The different pep−GO systems induce different effects of perturbation to the lipid membrane. In particular, for BDNF(1-12)−GO (Figure 4b), particles float on the membrane as if they were ‘surfing’ on it; on the other hand, the membrane seems to be highly perturbated by NT3 (1-13)−GO (Figure 3b) or NGF(1-14)−GO (Figure 4c).

The diffusion coefficients, obtained from the FRAP analysis for the pep−GO constructs after the interaction with the model 2D cell membrane (Table 5), confirm that both BDNF(1-12)−GO and GO do not significantly modify the lateral diffusion properties of the membrane, with the calculated diffusion coefficient being similar to that measured for the SLB as prepared (*D*~2.7 μm^2^·s^−1^), which is consistent with a homogeneous and fluid SLB. On the other hand, both NT3(1-13)−GO and NGF(1-14)−GO significantly change the membrane fluidity, with the calculated average diffusion coefficients being, respectively, higher and lower than the reference control SLB.

The analysis of the fluorescence curves for the three pep−GO materials at the interface with the 2D artificial cell membrane, as reported in Table 5, indicates a 75% value for the mobile fraction of molecules in SLB treated with GO. This value increases to around 82% for SLB treated with BDNF(1-12)−GO, while it decreases for the membranes treated with the other two peptides, especially in the case of NT3 (1-13)−GO. These results may be explained by the fact that, as discussed in the MD simulations, NGF(1-14) and NT3(1-13) have longer amino acid sequences than BDNF(1-12); both contain two aromatic residues, not present in BDNF, which are able to increase hydrophobic interactions: these promote the insertion of the pep−GO inside the lipidic bilayer.

Furthermore, the physisorption of the peptide chain on GO involves amino acids with positively charged side chains, such as arginine, in the middle of the BDNF(1-12) sequence, while both NGF(1-14) and NT3(1-13) display the enhanced polar character only in the N-terminal region (see Figure 2 above), thus giving a more head–tail oriented adsorption on GO, which is different to the side adsorption given by BDNF(1-12). This disposition will then leave the hydrophobic aromatic side groups freer and more available for interaction with lipid hydrophobic tails, promoting a deeper insertion of the peptide and the whole pep−GO inside the bilayer.

### 3.2. The Interaction of Peptide-Functionalized GO with Cells

#### 3.2.1. Cytotoxicity of NT Peptide-Functionalized GO in SH-SY5Y and HUVECs

The results of cell viability using MTT assays for the treatment with the pep−GO systems in the three cellular models considered in this study are shown in Figure 5.

Concerning HUVECs, Figure 5a indicates a small decrease in cell viability (~25% less viable cells than untreated control) for GO at the highest concentrations used in the experiment, namely, 5 μg/mL. This finding is in agreement with previous literature reports, where significant toxicity was found in HUVECs only at higher GO concentrations (25 and 50 μg/mL) [78]. The same trend observed in GO, i.e., a dose-dependent small decrease in cell viability, is maintained in the pep−GO hybrids with BDNF(1-12) and NT3(1-13). Noteworthy, in the case of the hybrid of GO with NGF(1-14), the opposite trend is found, and the statistically significant reduction in cell viability observed for the peptide alone is nullified for the treatment with NGF(1-14)−GO.

The cellular uptake of GO is a complex phenomenon affected by many factors, including the size of the particle and modifications to its surface. Notably, recent studies on GO nanosheets incubated with human serum found enhanced cell proliferation, but lesser efficiency in the cellular uptake than their bare nanosheet counterparts [79]. Moreover, the high or low surface oxygen content, such as in the respective cases of GO and rGO, induce the predominant parallel or perpendicular interaction with the cell membrane [80]. In our case, the functionalization of GO with NGF(1-14) seems to be the most effective in switching the interaction mechanism at the GO-cell membrane interface.

Concerning neuroblastoma, no evidence of significant toxicity for all the treatment conditions is found in undifferentiated cells (SH-SY5Y) (Figure 5b), except in the case of GO, which is known, even at low concentrations, to increase ROS production and induce autophagy in neuroblastoma SH-SY5Y cell lines [81,82].

On the other hand, for differentiated neuroblastoma cells (dSH-SY5Y), Figure 5c shows a proliferative effect of GO, especially at the lowest concentration tested. This finding is in agreement with previous reports on GO nanocomposites with silver nanoparticles, which were able to stimulate the differentiation of SH-SY5Y cells [83].

A similar activity is also maintained in the hybrid nanoplatform with BDNF(1-12), which instead, as a free peptide, does not induce any significant change in cell viability.

Notably, for both NT3(1-13) and NGF(1-14), which have a significant proliferative activity on dSH-SY5Y cells, the corresponding pep−GO hybrids exhibit the opposite trend, namely, a dose-dependent decrease in cell viability.

#### 3.2.2. Neuroblastoma and Neuron-like Cells (Neurite Outgrowth)

The neuronal migration and appropriate establishment of circuitry are extremely important processes in nervous system functioning [84]. During their development, neurons extend numerous processes (also called neurites) that differentiate into dendrites and axons, and are critical for communication between neurons [85]. Understanding the regulatory mechanisms which govern neuronal outgrowth, both during development and regeneration after an injury, is, thus, crucial for the development of treatments for spinal cord injury and neuropathological disorders; the investigation of these mechanisms strongly depends on in vitro analyses, which are important in identifying inhibiting or promoting factors in neuronal extension [86]. An example of such factors can be represented by the neurotrophins family [87]. The most commonly used technique to assess the ability of a factor to inhibit or promote neuronal growth relies on the measurement of L, which represents the length of the outgrowths [88,89,90]. The results of the optical micrograph analysis for SH-SY5Y and dSH-SY5Y neuroblastoma cells treated with the pep−GO systems are reported in Figure 6.

The NT3(1-13) sequence appears to be the most effective, either after 15 or 120 min, in aiding the growth of neurites when immobilized on GO sheets. On the other hand, BDNF(1-12)−GO showed a decrease in the mean length, which can probably be attributed to the fact that this peptide sequence gives the weakest interaction with GO sheets, as seen from FRAP and FRET experiments and molecular dynamics simulation, while NGF(1-14) gives an intermediate result between the previous two experiments.

Furthermore, undifferentiated SH-SY5Y cells show fewer and shorter neurites than differentiated cells, with a mean length that is about 82% of that shown through differentiated cells. To be noted is the effect induced using graphene oxide in undifferentiated cells: previous literature works effectively showed, for GO, an enhancing effect in neural cell differentiation [91]. In particular, GO can significantly enhance the differentiation of SH-SY5Y-induced retinoic acid [92]. 

Another very interesting piece of evidence from Figure 6 is that, in the short-term interaction (t = 15 min), both cell types exhibit a peptide-driven response upon treatment with the pep−GO hybrids in the membrane perturbation that, in turn, affects the neurite outgrowth process. On the other hand, in the long-term interaction (t = 120 min), the graphs display a trend for pep−GO samples where a major contribution from the GO surface, instead of the peptide ‘surface’, can be ruled out.

#### 3.2.3. Endothelial Cells (Angiogenesis Assays)

The intrinsic ability of peripheral nerves to regenerate after an injury is extremely limited, especially in the case of severe injury. Current approaches for the treatment of injured nerves often fail to support the survival and growth of nerve cells [93].

To exploit the angiogenic properties of our pep−GO systems, the stimulation of the tube formation and migration of HUVECs in vitro was assessed using the Matrigel and scratch assays, respectively

The results of the Matrigel test (Figure 7) point to a general triggering activity from both the neurotrophic peptides and the pep−GO systems. The effect is especially evident for the number of branching points after 5 h of incubation time with each of the three pep−GO samples. Notably, for the bare GO, we do not observe a statistically significant enhancement in the tube formation. This finding is not surprising, as GO is known to exhibit antiangiogenic activity, due both to the combination of the physical hindrance of the nanosheets aggregates internalized via the HUVECs, and the alteration of metabolic pathways, e.g., induced by oxidative stress [78].

Concerning the triggering effect, as determined via the wound scratch test, Figure 8 shows that the two shorter peptide sequences, namely, BDNF(1-12) and NT3(1-13), promote cell migration, both as free peptides and when immobilized onto the GO nanosheets.

In summary, in vitro cellular experiments on HUVECs and neuroblastoma, both undifferentiated and differentiated cells, pointed to the very promising potential of these pep−GO nanoplatforms to tune the interplay between neurogenesis and angiogenesis processes involved in neural regeneration. The three peptide sequences immobilized at the surface of the GO, each one with its own specificity in the sequence and charge, as well as in the average conformation in the adlayer onto the GO, resulted in triggering different effects in terms of cytotoxicity (or proliferative effect), neurite outgrowth, and angiogenesis from the matching of the intrinsic biological activities of the separate components, i.e., the free neurotrophic peptides and the bare GO nanosheets.

## 4. Conclusions

This work aimed to study the interface between the neurotrophin-mimicking peptides immobilized on the surface of GO nanosheets (pep−GO) and artificial models of cell membrane. Both the experiments and the computational analyses pointed out sequence-specific interactions for every studied neurotrophin-mimicking peptide and the graphene oxide nanoplatform. These findings identified a major role for electrostatic forces in triggering the average conformational structure of the peptide molecules in the adlayer, which in turn affect the interface with artificial phospholipid biomembranes, either in 3D (SUV) or 2D (SLB).

The in vitro cellular experiments on undifferentiated and differentiated neuroblastoma cells demonstrated the capability of the GO nanosheets functionalized with the three different peptide sequences to trigger the neurite outgrowth. Specifically, the comparative analyses for short- (15 min) or long-term (120 min) interaction times among SH-SY5Y, dSH-SY5Y, and the pep−GO hybrids evidenced trends pointing to the major role of the peptide or the GO substrate, respectively. The angiogenesis experiments showed that the pep−GO systems substantially stimulated the tube formation of HUVECs, while the cell migration promoting effect of BDNF(1-12) and NT3(1-13) peptides was maintained, both at the used experimental conditions and after their immobilization onto the GO nanosheets.

In conclusion, the developed systems are very promising as potential theranostic platforms to target neurodegenerative diseases and related vascularization processes.

Taken together, the neurotrophic and angiogenic properties of our pep−GO systems result are very promising for their triggering effect on the regeneration process.

## Data Availability

Data supporting reported results are not available online.

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
