# Peer review of "Bioinspired Nanoplatforms Based on Graphene Oxide and Neurotrophin-Mimicking Peptides"

_membranes, 2023, doi:10.3390/membranes13050489_

Round 1
Reviewer 1 Report
Luigi Redigolo et al. reported an interesting work about a biomimetic nanosystem with BDNF mimicking properties. The topic was interesting, and the manuscript fell within the scope of Membranes. The reviewer had several suggestions for improving this work:
1. The advantages of GO over other nanomaterials should be demonstrated. Now in the Introduction, it seemed that the authors “randomly” picked up GO.
2. It was better to state the reason why HUVECs were selected as the model cells. Maybe some CNS related cells would be more appropriate?
3. For Table 1, it was meaningless to give a MW on the scale of 0.0001, like 1680.7669. Perhaps 1681 or 1680.77 was okay.
4. The software used for MD should be named.
5. Despite of the hydrogen bond distance, it was suggested to also determine the hydrogen bond intensity or energy.
6. The scale bars in Figure 4 should be supplemented.
7. The cytotoxicity of developed system on HUVECs should be determined prior to other cellular tests.
8. Before the Conclusion Section, the aspects related with biomedical, clinical or industrial uses should be discussed.
Author Response
- The advantages of GO over other nanomaterials should be demonstrated. Now in the Introduction, it seemed that the authors “randomly” picked up GO.
Answer: we thank the reviewer for this comment. In the introduction section, the discussion on the advantages of graphene oxide in the context of this research has been extended with related references.
- It was better to state the reason why HUVECs were selected as the model cells. Maybe some CNS related cells would be more appropriate?
Answer: In this work, we used both CNS-related cells (undifferentiated and differentiated neuroblastoma cells) and angiogenesis-related cells (HUVEC), to compare the effects of our neurotrophin-functionalized nanomaterial on two cellular models able to rule out the tight relationship between angiogenic and neuronal signaling prompted by neurotrophins. This aspect has been expressed more explicitly in the introduction section of the revised manuscript.
- For Table 1, it was meaningless to give a MW on the scale of 0.0001, like 1680.7669. Perhaps 1681 or 1680.77 was okay.
Answer: The Table has been corrected, thanks.
- The software used for MD should be named.
Answer: We thank the Reviewer; the software name has been added in Section 2.5.3.
- Despite of the hydrogen bond distance, it was suggested to also determine the hydrogen bond intensity or energy.
Answer: We thank the Reviewer; the calculated H-bonds energies have been added.
- The scale bars in Figure 4 should be supplemented.
Answer: We thank the Reviewer; the scale bar has been added in the Results and Discussion section 3.
- The cytotoxicity of developed system on HUVECs should be determined prior to other cellular tests.
Answer: The results of cytotoxicity tests for both SHSY5Y and HUVECs have been added to the revised manuscript.
- Before the Conclusion Section, the aspects related with biomedical, clinical or industrial uses should be discussed.
Answer: The discussion section has been extended, before the conclusion section, to consider the Reviewer’s suggestion.
Reviewer 2 Report
membranes-2347118
Review of the article titled:
Bioinspired Nanoplatforms based on Graphene Oxide and Neurotrophin-Mimicking Peptides
by Luigi Redigolo, Vanessa Sanfilippo, Diego La Mendola, Giuseppe Forte, Cristina Satriano
In Membranes (ISSN 2077-0375).
Round 1
This study seems interesting. The experiments are well presented, and the results have value for practitioners. However, the discussion section is missed and incredibly minimal.
- Only four works have been cited since 2022 and one in 2023. Please provide a much more valuable reference list using references cited since 2023.
- Section 2.3.2. Please specify the dimension of the slit’s width (in nm) (The spectra were collected using 5:2.5 slit widths for all measurements.). Have you used degassed solution?
- Graphene oxide (GO) was purchased from Graphenea Inc., US. Please, provide any data concerning the purity of these sample, for example transition metal content. See as showed below:
Mikheev, I.V.; Byvsheva, S.M.; Sozarukova, M.M.; Kottsov, S.Y.; Proskurnina, E.V.; Proskurnin, M.A. High-Throughput Preparation of Uncontaminated Graphene-Oxide Aqueous Dispersions with Antioxidant Properties by Semi-Automated Diffusion Dialysis. Nanomaterials 2022, 12, 4159. https://doi.org/10.3390/nano12234159
The residual quantity of impurities may cause fluorescence quenching and showed another binding property.
- Section 2.3 Error! Reference source not found.
- Conclusion section should be expanded by future perspectives.
Author Response
- This study seems interesting. The experiments are well presented, and the results have value for practitioners. However, the discussion section is missed and incredibly minimal.
Answer: We thank the Reviewer for this comment. The discussion part has been extended.
- Only four works have been cited since 2022 and one in 2023. Please provide a much more valuable reference list using references cited since 2023.
Answer: The references have been implemented with more works from 2023.
- Section 2.3.2. Please specify the dimension of the slit’s width (in nm) (The spectra were collected using 5:2.5 slit widths for all measurements.). Have you used degassed solution?
Answer: We thank the referee for pointing this out. We corrected the issue in the text (now Section named 2.5.1). The PBS solution was filtered and degassed, as when used for the SUVs and SLBs preparation (see section 2.2).
- Graphene oxide (GO) was purchased from Graphenea Inc., US. Please, provide any data concerning the purity of these sample, for example transition metal content. See as showed below:
Mikheev, I.V.; Byvsheva, S.M.; Sozarukova, M.M.; Kottsov, S.Y.; Proskurnina, E.V.; Proskurnin, M.A. High-Throughput Preparation of Uncontaminated Graphene-Oxide Aqueous Dispersions with Antioxidant Properties by Semi-Automated Diffusion Dialysis. Nanomaterials 2022, 12, 4159. https://doi.org/10.3390/nano12234159
The residual quantity of impurities may cause fluorescence quenching and showed another binding property.
Answer: We acknowledge the reviewer’s note and thank them for highlighting it. We purchased a Graphene oxide water dispersion (0.4% wt% concentration), with GO particle size inferior to 10 μm; the producer’s elemental analysis reports the following composition: 49-56% C, 41-50% O, 2-3% S, 0-1% N, 1-2% H. This has been updated in section 2.1 in the text.
- Section 2.3 Error! Reference source not found.
Answer: The typo has been corrected.
- Conclusion section should be expanded by future perspectives.
Answer: The conclusion section has been expanded in the revised manuscript.
Round 2
Reviewer 1 Report
I have no further questions.
Reviewer 2 Report
I have no further concerns. The manuscript can be accepted.